# Humoral profiles of toddlers and young children following SARS-CoV-2 mRNA vaccination

Nadège Nziza[1,11], Yixiang Deng[1,2,11], Lianna Wood [1,3,11], Navneet Dhanoa [4], Naomi Dulit-Greenberg[4], Tina Chen [1], Abigail S. Kane[4,5], Zoe Swank[6,7], Jameson P. Davis [5], Melina Demokritou[4], Anagha P. Chitnis[5], Alessio Fasano [4,5,6], Andrea G. Edlow [6,8,9], Nitya Jain[4,5,6], Bruce H. Horwitz [6,10], Ryan P. McNamara[1], David R. Walt [6,7], Douglas A. Lauffenburger [2], Boris Julg [1,6], Wayne G. Shreffler [4,6], Galit Alter[1,6,12] & Lael M. Yonker [4,5,6,12] ✉

Although young children generally experience mild symptoms following infection with SARS-CoV-2, severe acute and long-term complications can occur. SARS-CoV-2 mRNA vaccines elicit robust immunoglobulin profiles in children ages 5 years and older, and in adults, corresponding with substantial protection against hospitalizations and severe disease. Whether similar immune responses and humoral protection can be observed in vaccinated infants and young children, who have a developing and vulnerable immune system, remains poorly understood. To study the impact of mRNA vaccination on the humoral immunity of infant, we use a system serology approach to comprehensively profile antibody responses in a cohort of children ages 6 months to 5 years who were vaccinated with the mRNA-1273 COVID-19 vaccine (25 µg). Responses are compared with vaccinated adults (100 µg), in addition to naturally infected toddlers and young children. Despite their lower vaccine dose, vaccinated toddlers elicit a functional antibody response as strong as adults, with higher antibody-dependent phagocytosis compared to adults, without report of side effects. Moreover, mRNA vaccination is associated with a higher IgG3-dependent humoral profile against SARS-CoV-2 compared to natural infection, supporting that mRNA vaccination is effective at eliciting a robust antibody response in toddlers and young children.

Despite the early misconception that children were spared from COVID-19, children continue to account for approximately twenty percent of all documented cases of COVID-19 infection in the United States, with infants and children under 5 years of age disproportionately affected by high rates of hospitalization[1]. While most children experience mild symptoms with acute SARS-CoV-2 infection, severe complications can ensue, even in the youngest children, and myocarditis, cardiomyopathy, renal failure, as well as coagulation and hemorrhagic disorders occur at increased rates with COVID-19[2]. Concerningly, COVID-19 deaths in children far exceed deaths from influenza[3] and COVID-19 is a leading morbidity and mortality in children in the United States[4].

SARS-CoV-2-targeting mRNA vaccines have become available for individuals six months of age and older[5–11]. These vaccines have

provided substantial protection against hospitalizations and severe disease in children ages 5–17 years[5,7,10,12]. Moreover, detailed humoral profiling of children and adolescents reveals that mRNA COVID-19 vaccines elicit robust, highly functional humoral immune responses in children in a dose-dependent manner[13], with strong cross-reactivity against variants of concerns (VOC)[13,14]. In children under 5 years of age, mRNA vaccination results in neutralizing immunoglobulin titers comparable to vaccinated adults and vaccination protects against symptomatic infection[11]. However, detailed humoral profiling in this age group has not yet been investigated. As an individual's humoral immune response evolves with age[15], age-related differences in mRNA vaccine responses must be fully characterized to fully understand the impact of mRNA vaccination in infants and toddlers.

In order to characterize the activation of humoral immunity in young children after SARS-CoV-2-specific mRNA vaccination, we used an unbiased system serology approach to analyze antibody levels and Fc-mediated functions in individuals ages 6 months through 5 years. We comprehensively profiled their antibody response following vaccination with the mRNA-1273 COVID-19 vaccine (25 µg) and compared it with antibody profiles of vaccinated adults (100 µg), as well as children infected with SARS-CoV-2. Our results reveal a strong activation of humoral immunity post-vaccination in these young children, with a highly functional and cross-reactive humoral immunity in comparison to adults and naturally infected infants.

## Results

### mRNA-vaccinated infants and toddlers generate robust Immunoglobulin G (IgG) responses

Our first objective was to profile vaccine-induced humoral immunity in infants and toddlers ages 0.6 through 5 years ($n = 18$) after completion of the two doses of the pediatric mRNA-1273 vaccination series (vaccine dose: 25 mcg mRNA-1273) and compare these serologic responses to those generated by fully vaccinated adults ($n = 13$; vaccine dose: 100 mcg mRNA-1273). In both groups, plasma samples were collected 2 months after the first vaccine dose (1 month after the second dose of the vaccine). Demographics of participants are included in Table 1;

**Table 1 | Demographics of mRNA-vaccinated and convalescent infants and children enrolled**

| Patient characteristics | mRNA-1273 vaccinated ($n = 19$) | COVID-Recovered ($n = 8$) | Total children enrolled ($n = 27$) |
|---|---|---|---|
| Age at Enrollment, mean (min, max) | 2.2 (0.6, 4.5) | 3.1 (1, 5) | 2.7 (0.6, 5) |
| Male Sex, number (%) | 7 (36.8) | 3 (38) | 10 (37) |
| Hispanic, number (%) | 8 (42.1) | 2 (25) | 10 (37) |
| Race, number (%) | | | |
| American Indian/ Native Alaskan | 1 (5.3) | 0 (0) | 1 (4) |
| Asian | 0 (0) | 1 (13) | 1 (4) |
| Black | 2 (10.5) | 0 (0) | 2 (7) |
| Other | 5 (26.3) | 2 (25) | 7 (26) |
| Unknown | 2 (10.5) | 0 (0) | 2 (7) |
| White | 9 (47.4) | 5 (63) | 14 (52) |
| Number of Samples per vaccine time point | | | |
| Pre vaccine (V0) | 14 | N/A | 14 |
| Post Vaccine #1 (V1) | 13 | N/A | 13 |
| Post Vaccine #2 (V2) | 9 | N/A | 9 |
| 6 months Post Vaccine #2 (V6) | 8 | N/A | 8 |
| Post Booster (VB) | 3 | N/A | 3 |

mean age of vaccinated pediatric participants was 2.2 years (range 7 months- 4.5 years). None of the vaccinated adults or children reported SARS-CoV-2 infections prior to or during their vaccine series, which was supported by the absence of elevated nucleocapsid responses (Fig. S1).

Our results demonstrate that despite their young age and receipt of only one quarter of the adult dose, total anti-Spike and anti-RBD IgG levels and IgG subclass in young children were similar to adults (Figs. 1A, S2). Interestingly, in contrast to IgG, this young population displayed lower levels of vaccine-induced anti-Spike and anti-RBD IgM and IgA1, which shows the distinct isotype selection between adults and children (Fig. 1A). We then compared the binding of spike and RBD-specific antibodies to Fc receptors (FcR), as well as antibody effector functions, including antibody-dependent cellular (monocyte) phagocytosis (ADCP), antibody-dependent neutrophil phagocytosis (ADNP) and antibody-dependent complement deposition or activation (ADCD) in young children and adults. We saw that infants and children less than 5 years old were able to produce antibodies with strong FcγR2A, FcγR2B, FcγR3A, and FcγR3B binding at similar levels as adults, and remarkably, anti-RBD antibodies exhibited stronger ADCP and ADNP effector functions in young children than in adults (Figs. 1B, C, S1). Antibodies from vaccinated young children displayed similar neutralization capacity as compared to vaccinated adults (Fig. S3).

To determine cross-reactivity of the vaccine-induced humoral response against variants of concerns (VOCs), we quantified antibody levels and FcR binding against Spike and RBD for six different SARS-CoV-2 VOCs including wild type (WT), Alpha, Beta, Gamma, Delta, and Omicron. While IgM, IgA1, and the FcR for IgA1 (FcαR) were higher in adults across the different SARS-CoV-2 variants, IgG response was essentially indistinguishable between young children and adults. In fact, the only exceptions were total IgG against RBD Omicron and IgG4 against Spike Gamma, RBD Alpha, RBD Delta, and RBD Omicron, which were significantly increased in young children (Fig. 1D).

To further characterize the capacity of the pediatric population to generate a broad SARS-CoV-2-specific humoral response following mRNA-1273 vaccination, we calculated a Spike and RBD protein breadth score. The breadth score highlights that infants and children less than 5 years old are able to induce a humoral response as robust as adults, with a strong recognition of different VOCs while IgM- and IgA-specific immunity is higher in adults (Fig. 1E). Taken together, these results show specificities regarding isotypes selection between young children and adults, with overall similar to enhanced antibody functionality against SARS-CoV-2 proteins in infants and toddlers less than 5 years old compared to adults.

When looking more broadly at antibody responses against common respiratory infections, including non-SARS-CoV-2 human coronavirus (HCoV) HKU1 Spike (HKU1), HCoV-OC43 Spike (OC43), and Influenza haemagglutinin (HA), we see a strong age-related difference. In contrast to the robust SARS-CoV-2 vaccine-induced humoral immunity across the age spectrum, young children have significantly lower antibodies titers against HKU1, OC43, and HA. Multivariate analysis highlights a clear separation between the two age categories distributions, as attested by the Partial least squares discriminant analysis (PLS-DA) (Fig. S5A). The LASSO-selected features that were used to build the PLS-DA model revealed an enrichment of antibody levels and FcγR binding against HKU1, OC43, and HA in adults (Fig. S5B). Co-correlates analysis showed strong connections between isotypes and FcγR features against non-SARS-CoV-2 antigens (Fig. S5C), all of which being enriched in older individuals. These antibody profiles in adults reflect prior exposure to these respiratory viruses over their lifetime, while these young children may remain naïve, particularly given the lower circulation of respiratory viruses observed during the COVID pandemic[16–18]. Alternatively, the lower level of antibodies could reflect lower total antigen-specific humoral

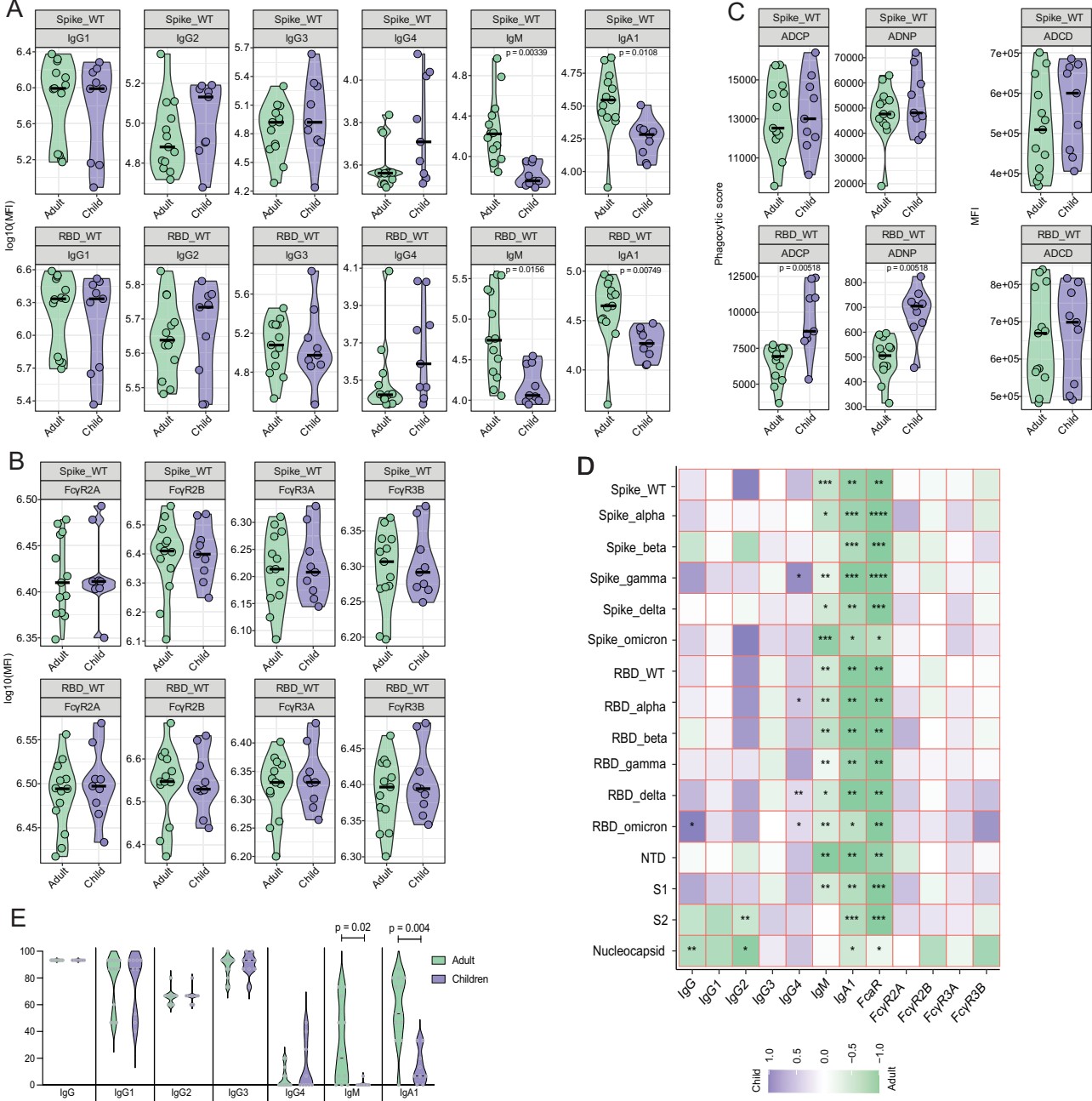

**Fig. 1 | mRNA-1273 vaccination induces a strong humoral immunity in children less than 5 years old.** Antibody levels and functionality were measured in the plasma of children less than 5 years old ($n = 9$; purple) and adults ($n = 13$, green), 2 months after vaccination. **A** IgG1, IgG2, IgG4, IgG4, IgM and IgA1 against Spike and RBD WT. **B** FcγR2A, FcγR2B, FcγR3A and FcγR3B binding against Spike and RBD WT. **C** Antibody-dependent cellular phagocytosis (ADCP), antibody-dependent neutrophil phagocytosis (ADNP), and antibody-dependent complement deposition (ADCD) against Spike and RBD WT. **D** Heatmap shows the univariate comparison between SARS-CoV-2-specific antibody response in children and adults. Difference

between the median of Z-scored MFI data are represented, where the color corresponds to the group that has the highest antibody response. **E** Breadth score was calculated by categorizing each antigen response as positive or negative, with positive response defined as 6 standard deviations above the mean of the COVID-unexposed controls, then calculating the percentage of positive Spike and RBD variant antigen responses for each secondary. Non-parametric two-sided Mann-Whitney U-test was used to calculate statistical significance, followed by Benjamini-Hochberg correction for multiple testing. *$p < 0.05$, **$p < 0.01$, ***$p < 0.001$, ****$p < 0.0001$. Source data are provided as a Source Data file.

responses to prior infection or the non-mRNA influenza vaccine, or more rapidly waning immunity in these young children.

### mRNA-1273 vaccination induces lasting, cross-reactive immunity in young children

In order to evaluate the impact of mRNA-1273 vaccination on the evolution of humoral immunity in young children, we measured antibody levels and Fc functionality prior to vaccination (V0), one month

after the first dose (V1), one month after the second dose (V2), six months after vaccination (V6), in addition to one month after boosting (Post-boost) (Figs. 2, 3). After just one vaccine dose, strong production of IgG, IgM, and IgA against Spike WT could be observed in these infants and children (Fig. 2A), with robust antibody binding to FcγR (Fig. 2B). Similarly, antibody effector function, characterized by ADCP and ADCD, was significantly increased at V1 compared to V0 (Fig. 2C). Peak antibody responses were generally observed 2 months after the

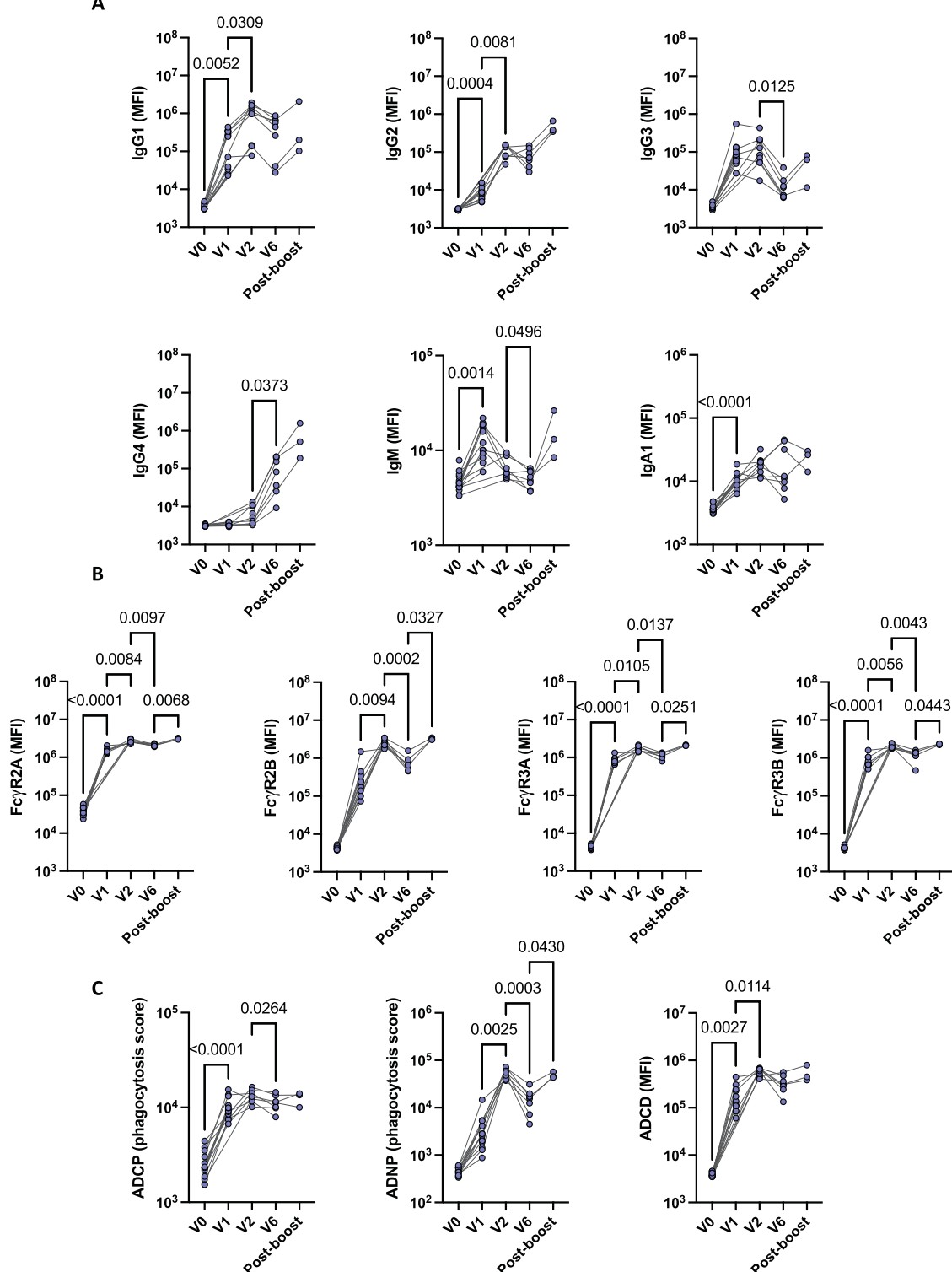

**Fig. 2 | Strong impact of mRNA-1273 vaccination on antibody response over-time.** Antibody levels **A**, binding to FcγR **B**, and function **C** against Spike WT was analyzed in children at different timepoints: before vaccination (V0, $n = 14$), 1 month (V1, $n = 13$) after the first dose of mRNA-1273 vaccine, 2 months (V2, $n = 9$) after the first dose of mRNA-1273 vaccine, 6 months (V6, $n = 8$) after the first mRNA-1273 vaccine series, as well as 1 month after boosting (post-boost, $n = 3$). Connecting lines represent identical individuals that were followed over time, and statistical differences were calculated between 2 consecutive timepoints. Non-parametric, two-sided Wilcoxon signed rank test was used to calculate differences between timepoints for paired data, followed by Benjamini-Hochberg correction for multiple testing. *P* values that are <0.05 are indicated on the graphs. ADCP antibody-dependent cellular phagocytosis, ADNP antibody-dependent neutrophil phagocytosis, ADCD antibody-dependent complement deposition. Source data are provided as a Source Data file.

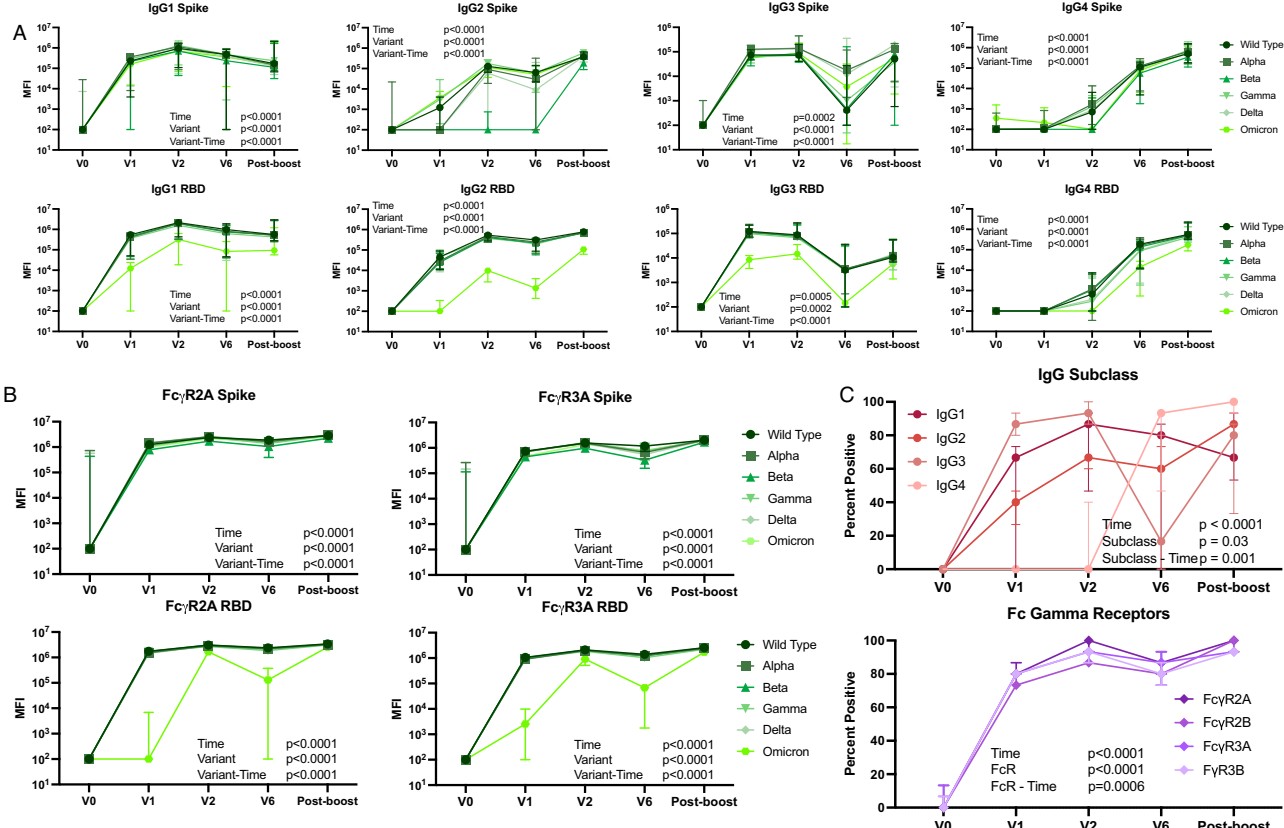

**Fig. 3 | Similar antibody responses across variant epitopes.** Median and 95% confidence interval at each timepoint for **A** Total IgG, IgG1, IgG2 and IgG3, as well as **B** FcγR2A and FcγR3A binding against Spike and the RBD. Differences in variant responses were tested by mixed effects model with Geisser-Greenhouse correction with time and antibody features as the two effects that were modeled. **C** Breadth score was calculated by defining positive responses to each antigen response as 6 standard deviations above the mean of the COVID-unexposed controls for the same antigen and subclass or FcR. We then calculated the percentage of Spike and RBD variant antigen responses for each secondary individual at each timepoint. V0: $n = 14$; V1: $n = 13$; V2: $n = 9$; V6: $n = 8$. Source data are provided as a Source Data file.

first dose of vaccination (Fig. 2), as attested by the high antibody levels and functionality (Fig. 2C). Of note, IgM levels started to wane after V1 (Fig. 2A). In the setting of rising IgG and IgA1 titers, this supports antibody maturation with efficient class switching. Although the number of individuals that was included for the post-boost analysis was low, our results highlighted a strong activation of the immune system one month after boosting, especially for FcγR binding and antibody-induced neutrophil activation (Fig. 2B, C).

To evaluate the ability of mRNA-1273 vaccination to elicit broadly cross-reactive antibody responses and their durability over time in young children, we compared the antibody responses to Spike antigens from wild type through Omicron variants at each time point (Fig. 3). Total IgG responses to full-length Spike were similar across all variants. However, IgG responses to the Omicron RBD were consistently lower post-vaccination for all subclasses and FcγRs (Fig. 3A, B), which is not unexpected given the large number of mutations in the RBD of Omicron in comparison to other variants and is consistent with cross-reactivity seen in older individuals[13,14]. To determine the breadth of antibody responses over time, breadth scores were calculated for each IgG subclass and each FcγR over the time as described above (Fig. 3C). Breadth was highest for IgG3, although this response did wane prior to boost. IgG2 and IgG3 responses both expanded with boosting, with minimal change in IgG1 responses. FcγR binding showed similar breadth for each FcγR tested, with a robust initial response, some waning in response at 6 months after vaccination, and increased breadth after boosting. Again, the breadth scores highlighted a broad anti-SARS-CoV-2 antibody response shortly after

vaccination that wanes over 6 months, but then appears to re-expand to peak levels post-boost.

## Vaccination produces greater IgG3 than natural infection in young children

To evaluate whether natural infection induces equivalent immunity compared to vaccination, we compared anti-Spike and anti-RBD titers, and Fc binding and effector function in serum collected from a group of 8 children (Table 1: mean age, 3.7 years; range: 1–5 years) one-month following acute SARS-CoV-2 infection, defined as symptomatic COVID-19 confirmed by PCR or rapid antigen test at the time of illness, and a second group of children one month after completion of their first dose of vaccine (V1). We did not detect a significant difference in total IgG levels (Fig. 4A), FcγR binding (Fig. 4B), and antibody functionality (Fig. 4C) between the naturally infected group and the vaccinated group at this one month timepoint (Fig. S4). This suggests that the induction of humoral immunity following vaccination is as strong as the response induced by natural SARS-CoV-2 infection in infants and toddlers. Interestingly, the IgG3 response to both Spike and RBD was significantly higher in vaccinated young children compared to infected children (Fig. 4A). Levels of IgG3, a highly potent IgG subclass with the greatest levels of functionality[19–21], correlate strongly with neutralization[22,23], suggesting that in young children vaccination induces a more mature and potent antibody response than natural infection with SARS-CoV-2. Further, this robust vaccine-induced IgG3 response is consistently elevated across different VOCs (Fig. 4D) highlighting the benefit of cross-reactivity gained by vaccination

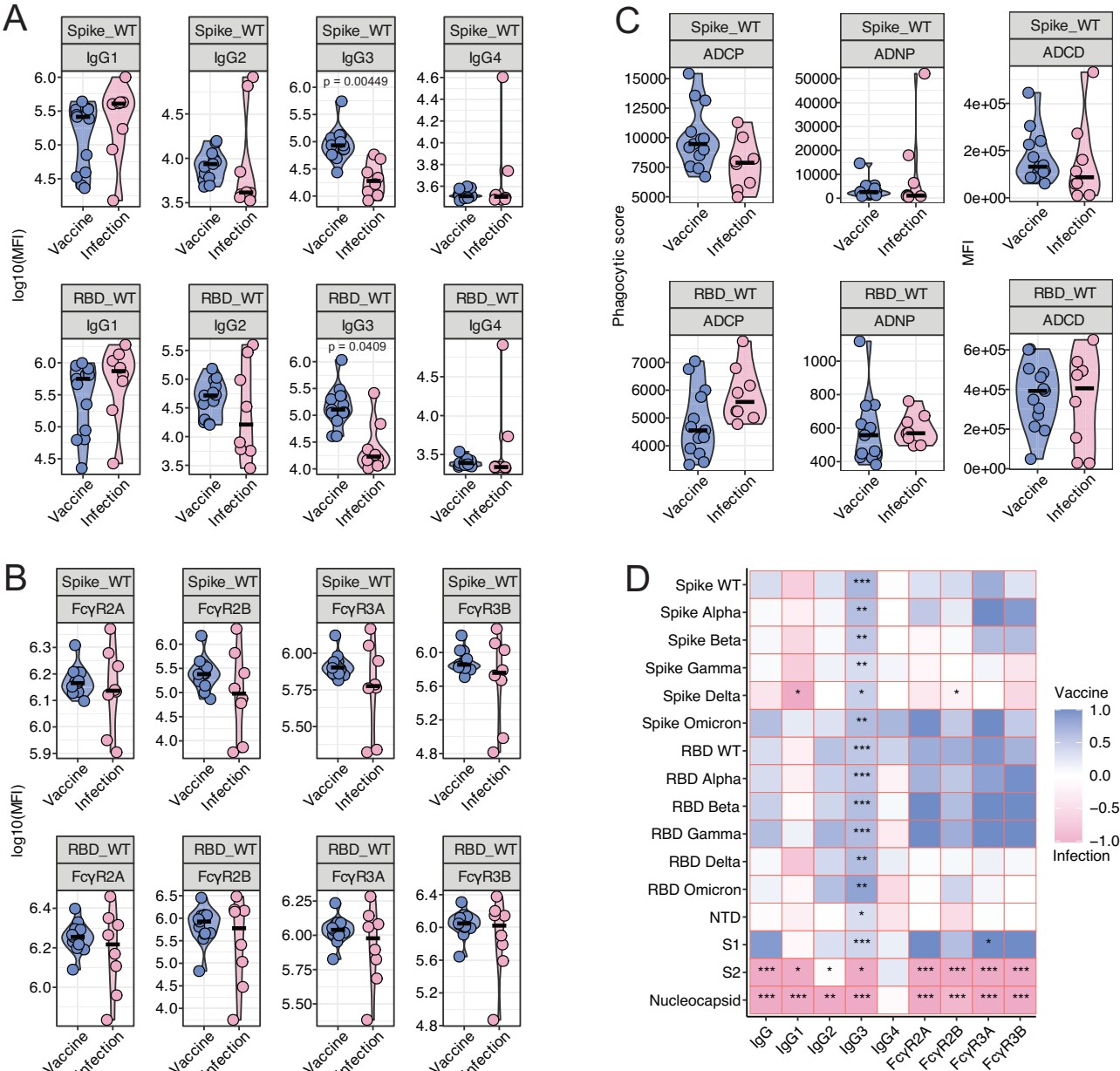

**Fig. 4 | Distinct antibody response between vaccination and natural infection in children under five.** Antibody response was measured in the plasma of children 1 month after vaccination ($n = 13$; blue) or diagnosis of COVID-19 ($n = 8$; pink). **A** IgG1, IgG2, IgG4 and IgG4 against Spike and RBD WT. **B** FcγR2A, FcγR2B, FcγR3A and FcγR3B binding against Spike and RBD WT. **C** Antibody-dependent cellular phagocytosis (ADCP), antibody-dependent neutrophil phagocytosis (ADNP), and antibody-dependent complement deposition (ADCD) against Spike and RBD WT. **D** Heatmap illustrates the difference between the median of Z-scored MFI data of the vaccinated and infected groups. Non-parametric two-sided Mann-Whitney U-test was used to calculate statistical significance, followed by Benjamini-Hochberg correction for multiple testing. *$p < 0.05$, **$p < 0.01$, ***$p < 0.001$. Source data are provided as a Source Data file.

compared to natural infection. As expected, anti-nucleocapsid antibody responses induced by natural infection were absent in children vaccinated with the mRNA-1273 vaccine as the vaccine does not encode for nucleocapsid (Fig. 4D). Of note, the elevated IgG1 levels against the Delta strain of Spike in the infected group reflect the VOC at time of infection based on the timing of samples collected (eight of ten of the children were infected with SARS-CoV-2 prior to the Omicron surge) highlighting the specificity of response in natural infection. Notably, vaccine-induced IgG3 levels still remained significantly higher than natural infection, even for the Delta strain. Taken together, these results show that mRNA-1273 vaccination in infants and children less than 5 years of age elicits strong humoral activation, with production of a highly mature and developed antibody response, suggesting a

more effective humoral response following vaccination in comparison to natural infection with SARS-CoV-2.

## Discussion

The availability of novel mRNA vaccine technology represented a key inflection point in the COVID-19 pandemic, dramatically reducing hospitalizations and deaths caused by SARS-CoV-2. However, with the novelty of the mRNA vaccine strategies, the impact on immune response in pediatric populations remains largely unknown, while several studies have described vaccine-dependent humoral activation in adults[24–26]. Thus, the risk/benefit ratio of vaccinating young children must be thoroughly analyzed; comprehensive profiling of the humoral immune response following vaccination, including characterization of

antibody response profiles and cross-protective potential is critical. While in-depth antibody titers and effector function have been described for mRNA-vaccinated adults and children ages 5 years and older[13,14], limited information exists for younger age groups.

Humoral responses are known to vary with age[27], with the capacity to generate antibodies increasing over time, including following administration of SARS-CoV-2 mRNA vaccines[13,14]. Remarkably, despite receiving only a quarter of the adult dose, our study suggests that infants and toddlers younger than 5 years are capable of generating titers of anti-SARS-CoV-2 IgG that are comparable to adults. The impact of diminished IgA and IgM in children in the setting of SARS-CoV-2 is not clear. In adults with COVID-19, elevated levels of anti-SARS-CoV-2 IgA immune complexes are associated with severe disease[28] and in children with MIS-C, anti-SARS-CoV-2 IgA immune complexes activate intravascular neutrophil extracellular traps which may contribute to endothelial damage[29]. Circulating IgA does not directly correlate with mucosal IgA[30,31] and in this study, we did not test for presence of antibodies at the mucosal surface. Future studies will be needed to fully characterize mucosal immunity following vaccination and to compare mucosal responses in children and adults.

Fc binding capacity of anti-SARS-CoV-2 IgG may play important protective functions including enhanced activation of monocytes and neutrophils. Here we demonstrated that vaccinated young children display comparable Fc binding capacity as compared to vaccinated adults, while significantly higher antibody functionality was observed in the younger population in comparison to adults, showing a potential impact of age-dependent antibody glycosylation on the induction of phagocytosis[32]. The pediatric nasal passages contain higher quantities of neutrophils than adults and these pediatric neutrophils tend to be primed for anti-viral responses[33]. Thus, the combination of these activating antibodies and primed neutrophils may lead to efficient containment of the virus at the nasal surface. Direct humoral profiling of the pediatric mucosal surface may reveal important differences between children and adults with potential implications for current vaccine strategies, as well as the development of mucosal vaccination strategies for COVID-19.

In addition to this strong and functional antibody response in young children two months after the first dose of vaccination, our results showed that this pediatric population was able to maintain functional humoral immunity for at least 6 months. We observed signs of efficient antibody class switching[34,35], as IgM levels rapidly decreased 1 month after vaccination, when IgG and IgA continued to be produced, in addition to increasing FcR binding and Fc-mediated functionality. Moreover, the analysis of vaccine-induced humoral immunity against VOC highlighted a strong and sustained antibody response over time with Alpha, Beta, Gamma, and Delta, while Omicron-specific immunity tended to be slightly lower, as reported previously[13]. It has been hypothesized that the naïve pediatric immune system facilitates the evolution and adaptation of immune response to allow broader immunity against future viral exposures[36,37], which might explain the more robust VOCs-specific antibody response in infants compared to adults. Of the IgG subclasses, though, IgG3 declined the most by 6 months but responded well to boosting, highlighting the importance of boosters in maintaining effective protection against SARS-CoV-2 over time. Collectively, these data suggest that the vaccine can provide long-term immunoprotection against COVID-19 in young children, with likely efficacy against emerging VOCs.

Studies in adults show that COVID-19 vaccination elicits a more robust antibody response compared to infection[38–40], with higher FcR binding capacity and functionality[25]. In the young cohort described in this project, the main difference observed between infection and vaccination at the one month timepoint included higher IgG3 levels against different VOCs after vaccination. With IgG3 being the most functional IgG subclass[19–21,41,42], these data show that vaccination in this young population elicits a stronger and potentially more functional

humoral immune response compared to natural infection. Additional analyses with larger cohorts would be needed to further characterize the age-dependent clinical impact of vaccination versus infection on antibody functionality, particularly with the study of antibody glycosylation profile, which is known to play a key role in the modulation of antibody functionality. Moreover, to address the question regarding which group is associated with superior clinical protection against COVID-19, further studies involving pharmacokinetic aspects, such as IgG3 half-life and the stability of this isotype in the blood of this pediatric population, as well as correlations between clinical features and antibody functionality, would be valuable in determining whether mRNA vaccination confers superior protection than SARS-CoV-2 infection. We also observed that the antibody response against Delta, which is the strain that was circulating at the time of sample collection, was higher in the infected group. This suggests that adapting vaccine strategies to incorporate genetic variations that appear in emerging respiratory viruses will be an important strategy to maintain vaccine efficacy.

While our study is limited in size, the overall population of vaccinated young children is limited in part because of parental/guardian vaccine hesitancy, as well as the current reduced uptake of the COVID vaccine following the relaxation of restrictions. Additionally, routine phlebotomy presents numerous challenges in children. However, our data set advances the depth of understanding of antibody responses to the mRNA vaccine in young vaccine-eligible children and helps inform the risk/benefit ratio for providers and parents/guardians. As vaccines result in a dramatic improvement in morbidity and mortality of adults related to COVID-19 following mRNA vaccination[43], and we see comparable- if not improved- vaccine responses in young children, we expect vaccines will also reduce severe disease and long-term complications in this young population as well. As COVID-19 has become one of the leading infectious causes of death in children, and infected children can suffer from post-COVID complications[44], vaccination strategies for these young children, as well as their impact on the maturing immune system, need to be studied in depth. Overall, our data suggest that vaccination offers robust protection against future SARS-CoV-2 infections, potentially superior to natural infection, and thus supports the notion that mRNA vaccination of this youngest group is highly effective. These results also provide insight into the design of future mRNA-based vaccine technologies for this pediatric population.

## Methods
### Participant enrollment
Families with children 6 months to 5 years of age who received the Moderna 25 µg mRNA1273 vaccine ($n = 13$) at Massachusetts General Hospital (MGH) were approached for enrollment in the MGH Pediatric COVID-19 Biorepository (IRB: 2020P000955) (Table 1). Vaccination schedule was performed in accordance with CDC guidelines. Families with children ($n = 8$) who were infected with SARS-CoV-2 in the past 5 weeks (average = 5.3 weeks ± 2 weeks), during the Delta wave, and were presenting to MGH for a well-child visit or hospital visit were also approached and offered enrollment in the MGH Pediatric COVID-19 Biorepository. Parents/guardians provided informed consent prior to participation. An adult cohort was also included in this project, which was described previously[45].

### Sample collection
Blood was collected prior to vaccination (Pre-vaccine), one month following the first vaccination (V1), one month following the second vaccination, (V2), and six months following the second vaccination (V6). If a booster dose was received, blood was collected prior to receipt of the booster dose (if greater than six months from first vaccination), and one month following the booster (post-boost). Participants could opt out of providing blood at any of the time points. Blood

was collected by venipuncture or by capillary microneedle device, processed for plasma, and stored at −80 °C. All procedures were approved by the MGB IRB.

Banked samples from adults who had received the Moderna mRNA1273 vaccine were used for comparison ($n = 13$). Detailed information regarding enrollment and specimen collection for this IRB-approved study were previously published[45].

## Antigens

SARS-CoV-2 D614G or variants of concern Spike and RBD proteins, in addition to NTD, S1, nucleocapsid, HCoV-HKU1 Spike (HKU1), and HCoV-OC43 Spike (OC43) antigens, were expressed in mammalian HEK293 cells and purchased from Sino Biological. Influenza haemagglutinin (HA) was obtained from Sino Biological. NHS-Sulfo-LC-LC kit was used for antigen biotinylating, according to the manufacturer's instruction (Thermo Fisher Scientific).

## Antibody isotype and FcR binding

Antibody isotype and subclass levels, as well as Fc-receptor (FcR) binding profiles were measured using a custom multiplex Luminex assay as described previously[46–48]. Briefly, Luminex microspheres (Luminex Corp) were coupled to antigens, then incubated with diluted plasma samples between (1:50 for the analysis of IgG2, IgG4, IgA1, IgM, and FcαR; 1:100 for the analysis of IgG, IgG1, and; 1:500 for the characterization of FcγR binding). After a 2-h incubation at room temperature, Ig isotype and subclasses were detected using phycoerythrin (PE)-conjugated secondary antibody at 1.3 µg/ml (mouse anti-human IgG, IgG1, IgG2, IgG3, IgG4, IgM, or IgA1 from Southern Biotech). Concerning FcR binding, PE–streptavidin (Agilent Technologies) was coupled to recombinant and biotinylated human FcR protein (FcγR2A, FcγR2B, FcγR3A, FcγR3B, and FcαR) purchased from Duke Human Vaccine Institute. After 1-h incubation at room temperature with either subclasses/isotypes or FcRs, immune complexes were washed and the median fluorescence intensity (MFI) of antibody levels or binding to FcRs was determined using an iQue analyzer (Intellicyt) (Fig. S6).

## Antibody-dependent complement deposition (ADCD)

Complement deposition was performed as described previously[49], using Luminex beads (Luminex Corp). Briefly, Luminex microspheres were coupled to biotinylated antigens, and immune complexes were formed using 1:30 diluted plasma samples. After a 2-h incubation at 37 °C, guinea pig complement in GVB++ buffer (Boston BioProducts) was added to immune complexes for 20 min at 37 °C. To stop the complement reaction, EDTA-containing phosphate-buffered saline (15 mM) was used, then C3 deposition on beads was detected using a 1:100 diluted anti-guinea pig complement C3 antibody (MP Biomedicals). MFI values were analyzed by flow cytometry on an iQue analyzer (Intellicyt) (Fig. S6).

## Antibody-dependent cellular phagocytosis (ADCP)

THP-1 monocytes (American Type Culture Collection) were used to determine ADCP, as previously described[50]. Briefly, FluoSphere NeutrAvidin beads (Thermo Fisher Scientific, $9 \times 10^5$ beads per well) were coupled to biotinylated antigens, followed by an incubation for 2 h 37 °C with 1:50 diluted plasma to form immune complexes. THP-1 (200 µl of cell suspension per well) was then added to the immune complexes, at a concentration of $1.25 \times 10^5$ cells/mL. After a 16-h incubation at 37 °C, THP-1 was fixed with 4% paraformaldehyde and then analyzed by flow cytometry on an iQue analyzer (Intellicyt) (Fig. S6).

## Antibody-dependent neutrophil phagocytosis (ADNP)

For ADNP, primary human neutrophils were used as previously described[51]. Similarly to ADCP, immune complexes were formed with antigen-coupled neutravidin microspheres (Thermo Fisher Scientific, $9 \times 10^5$ beads per well) and antibodies from diluted plasma samples (dilution 1:50). After a 2-h incubation at 37 °C, neutrophils isolated from healthy donors' blood were added and incubated for 1 h at 37 °C, (200 µl, at a concentration of $2.5 \times 10^5$ cells/mL). Neutrophils were then surface stained with anti-human CD66b Pacific Blue antibody (BioLegend), fixed with 4% paraformaldehyde, and analyzed by flow cytometry on an iQue analyzer (Intellicyt) (Fig. S6).

## VOC breadth score

Spike and RBD protein breadth score were calculated by categorizing each antigen response as positive or negative and calculating the percentage of Spike and RBD variant antigen responses for each secondary (isotype or FcR) at each timepoint. We defined a positive response as six standard deviations above the mean of the SARS-CoV-2-unexposed controls for the same antigen and isotype or Fc receptor.

## Neutralization assay

Neutralization capacities were measured using previously developed bead-based competitive inhibition assays[52]. Recombinant SARS-CoV-2 spike protein was conjugated to 647 nm dye-encoded magnetic beads to measure neutralization capacities against the wild-type strain, and recombinant omicron spike was conjugated to 750 nm dye-encoded magnetic beads to measure neutralization capacities against the omicron strain. To validate our reagents, we incubated the spike-coated beads with biotinylated angiotensin-converting enzyme 2 (ACE2) and two neutralizing antibodies against the wild-type (40592-R001, Sino Biological) and the omicron (40592-MM117, Sino Biological) spike proteins. As the neutralizing antibody concentration increases, the assay signal decreases to background level, similarly for each variant.

## Statistical analysis

All experiments were done in duplicate and the analysis was conducted using the average of the duplicates. Phosphate-buffered saline (PBS) and plasma from healthy donors were used as negative controls. GraphPad Prism (v.9.2.0) and RStudio (v.1.3 and R v.4.0) were used to perform data analyses. We calculated the breadth score by categorizing each antigen response as positive or negative and calculating the percentage of positive Spike and RBD variant antigen responses for each antibody feature at each timepoint. We defined a positive response as six standard deviations above the mean of the COVID-unexposed controls for the same antigen and isotype or Fc receptor. Differences in variant responses were tested by a mixed effects model with Geisser-Greenhouse correction with time and antibody features as the two effects that were modeled (Fig. 3).

Multivariate analyses to compare vaccinated adults and children were built as described previously[50,53]. Data were normalized using z-scoring, and then a least absolute shrinkage and selection operator (LASSO) approach was used for feature selection. For classification and visualization, partial least square discriminant analysis (PLS-DA) models were performed using LASSO-selected features, followed by a ten-fold cross-validation to assess model accuracy. Co-correlates of LASSO selected features were represented in a network format and identified using the Spearman method followed by Benjamini-Hochberg correction.

## Reporting summary

Further information on research design is available in the Nature Portfolio Reporting Summary linked to this article.

# Data availability

All relevant data are included in this manuscript in the Source Data file. Source data are provided in this paper.

## Code availability

There was no specific custom code used in this manuscript. All code is publicly available, and the source is indicated in the text and/or the Methods section. Scripts will be made available upon reasonable request.

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

## Acknowledgements

We thank N. Zimmerman, T. Ragon, S. Ragon, L. Schwartz, M. Schwartz, and the SAMANA Kay MGH Research Scholars award for support. We acknowledge support from the Ragon Institute of MGH, MIT, and Harvard, Gates Foundation Global Health Vaccine Accelerator Platform funding (OPP1146996 and INV-001650 to G.A.) and the NIH (3R37AI080289-11S1 to G.A., R01AI146785 to G.A., U19AI42790-01 to G.A., U19AI135995-02 to G.A., U19AI42790-01 to G.A., 1U01CA260476-01 to G.A., CIVIC75N93019C00052 to G.A., 5K08HL143183 to L.M.Y.).

## Author contributions

N.N. and T.C. performed the serological experiments. N.N., Y.D., L.W., N.D., N.D.G., R.P.M., B.J., D.A.L., B.J., R.P.M., D.R.W. G.A. and L.Y. analyzed and interpreted the data. A.S.K., Z.S., J.P.D., M.D., A.C., A.F., A.E., N.J., B.H., W.S. and L.Y. managed the sample collection. N.N., G.A. and L.Y. drafted the manuscript. All authors critically reviewed the manuscript.

## Competing interests

G.A. is a V.P. at Moderna, a founder and equity holder of Seromyx Systems, and an employee and equity holder of Leyden Labs. G.A.'s interests were reviewed and are managed by MGH and Partners HealthCare by their conflict-of-interest policies. All other authors declare that they have no competing interests.

## Additional information

[1]Ragon Institute of MGH, MIT, and Harvard, Cambridge, MA, USA. [2]Department of Biological Engineering, Massachusetts Institute of Technology, Cambridge, MA, USA. [3]Boston Children's Hospital, Department of Pediatric Gastroenterology, Boston, MA, USA. [4]Massachusetts General Hospital, Department of Pediatrics, Boston, MA, USA. [5]Massachusetts General Hospital, Mucosal Immunology and Biology Research Center, Boston, MA, USA. [6]Harvard Medical School, Boston, MA, USA. [7]Department of Pathology, Brigham and Women's Hospital, Boston, MA, USA. [8]Massachusetts General Hospital, Department of Obstetrics and Gynecology, Division of Maternal-Fetal Medicine, Boston, MA, USA. [9]Massachusetts General Hospital, Vincent Center for Reproductive Biology, Boston, MA, USA. [10]Boston Children's Hospital, Department of Emergency Medicine, Boston, MA, USA. [11]These authors contributed equally: Nadège Nziza, Yixiang Deng, Lianna Wood. [12]These authors jointly supervised this work: Galit Alter, Lael M. Yonker. ✉e-mail: Lyonker@mgh.harvard.edu

