## [Peer Review File · Nature Communications]

Humoral profiles of toddlers and young children following SARS-CoV-2 mRNA vaccinationReviewers' Comments:

Reviewer #1:

Remarks to the Author:

This manuscript by Nziza, Deng, Wood and colleagues describes the antibody profiles and function in young children less than 5 years of age following mRNA vaccination using systems serology. Responses were compared with adults as well as age-matched children infected with SARS-CoV-2. They report that children mount robust humoral antibody responses following vaccination (as expected) but that this response differed in some aspects when compared with adults or children who were infected. In particular, vaccinated children were found to have lower Spike and RBD-specific IgM and IgA1 responses compared with adults as well as stronger ADCP and ADNP effector functions. The authors also found some differences in the breadth of the response in children as well as responses to non-SARS-CoV-2 respiratory viruses compared with adults. Data was also presented over time (out to 6m following primary series, and following booster vaccination), showing waning of humoral immunity by 6 months but which was restored by the booster. Importantly, an IgG3 signature was identified in vaccinated children compared with infection, which provides novel evidence for the benefit of vaccinating children.

This is a very interesting manuscript that provides detailed information on the humoral immune response in young children following mRNA vaccination. This is the paper's strength since very limited data from young children have been reported in this context. I do however have a number of points that need to be considered by the authors.

1. The major limitation of this manuscript, as acknowledged by the authors, is the small samples size. Notwithstanding the variety of analyses conducted on these samples, the small numbers does make it difficult to draw meaningful conclusions. In particular, for some comparisons (e.g. timecourse Fig 2) there are <10 individuals. There are no sample size calculations provided which would provide confidence in these data.
2. Table 1 needs to include the number of samples studied at each timepoint as well as the time post vaccination or infection that the samples were measured. In the text, there are inconsistencies from 2-8 weeks post-infection and 1 month post. This could have an impact on the responses observed if only 2 weeks post-infection. If so, have the authors considered any analyses to account for time since infection/vaccination on the response observed?
3. A statement in the methods on the time period that these samples were collected. It appears that these were collected during the Delta wave but this is not made clear.
4. For the cross-reactivity analyses with other respiratory infections (supplementary data), the authors report lower responses in children compared with adults to these antigens. This is consistent with the literature but given that the circulation of respiratory viruses were very low during the time that these samples were collected, could this be another explanation for the differences observed?
5. The authors show data in young children <5years following a booster dose of mRNA vaccination. Is this an approved schedule in the US as I thought booster vaccinations were only recommended for children over 5 years of age? This needs to be clarified in the methods.
6. In some Figures, there is data on all IgG subclasses (Figs 1 and 4) but not others (Figs 2 and 3). Was this done for all analyses?
7. In Figure 1, it shows no difference in the FcγR response between children and adults, yet there was a difference in ADNP observed? What do the authors think might be the explanation for this?
8. Fig 4 legend states "...plasma of children 1 month after vaccination or diagnosis (on average)" – the authors should provide the precise numbers for this, ie mean and range or similar
9. For the breadth score (Fig 3), it states that a COVID-unexposed group was used to control for the response. Was this an age-matched control group? It is also not clear what the p-values are referring to on these graphs – are these global changes? It appears that this difference is largely attributed to the Omicron response which is not surprising particularly as this was mainly seen pre-booster.

Reviewer #2:

Remarks to the Author:

Dear Editor,

I have read the manuscript entitled "Humoral profiles of toddlers and young children following SARS-CoV-2 mRNA vaccination" by Nziza et al.

They have performed a study of 19 mRNA-1273 vaccinated (2-doses) children under the age of 5 (25 mcg), and compared these to 8 naturally infected children. They also compared with an undisclosed group of 13 fully vaccinated (3-4 doses?) adults that had received 100 mcg of the vaccine. They find that the vaccinated children generated a robust IgG response, compared to adults, except for IgA and IgM levels, which are lower. They make strong statements about the benefits of vaccination compared to natural infection and that children have stronger functional responses than adults, which in my view, are not supported by the data shown.

The study has the potential to shed some light on the vaccination of young children, given that they have access to such samples, but the study design, experiments, and data presentation are severely lacking.

The results that appear to be clear are that young children get a good IgG response but weaker IgA and IgM response and that IgG3 levels might (pending on how the experiments were made, which is not clear) differ between vaccinated and infected children.

Major points:

1. Almost non-existent methods description. They very briefly refer to a Luminex-based method with no details. The reference they point to only studies subclass, and it would be impossible to reproduce subclass analysis given that they don't reveal any details. In addition, they state that they also measure IgA1, IgM, and FcγR2A, FcγR2B, FcγR3A, FcγR3B, and FcαR, none of which are described or measured in the cited reference.

Similarly, they report that they measure ADCP, ADCC, ADNP with no details. Only refer to other papers of the group, where it is impossible to find information specific to this study. Which cells do they use? At what concentration? Volume? Beads? Antigens? Time? The ratio of beads to cells? Titration curves? What controls did they use for the methods? etc. They write that they record MFI values but no flow cytometry details such as gatings, settings etc. How many times were the experiments repeated? This is really sub-standard and makes it impossible to assess the quality of the data.

Fig 1 and 4. ADCP, ADCC, ADNP log values of what? No description of what these values represent. One would have to guess. Which makes it impossible to evaluate. How many times were these analyses made? 1 time?

How were the heat maps constructed in Fig 1 and 4? From how many experiments?

2. In line 86-88 the authors stress that the anti-spike IgG levels are similar to adults. The comparison of the antibody response is to this adult cohort but the details concerning this cohort is limited. At what time point after vaccination were the IgG responses of the adult group compared? This needs to be clarified since it affects the interpretation (antibody titers wane over time) and can be a major bias in the interpretation of the results if the blood samples were taken several months after vaccination since it would affect titers and isotype presence etc. The novelty would have been greatly improved if the analysis of the vaccinated and unvaccinated cohort would entail adults (at the same time point of 1 month) and include IgA and IgM as well as neutralization assays.

3. The breadth score, found in graph 1E, shows that there is no difference between IgG3 levels for

adults or children under 5. However, the authors show that natural infection vs vaccination, the humoral immune response differs for children (Figure 4). However, this same comparison with adults is not shown. This analysis for a comparable cohort (vaccinated adults one month after or infection one month after) is essential to show to observe if there is an age-dependent IgG3-skewed response depending on infection or vaccination.

4. They describe “remarkably, anti-RBD antibodies exhibited stronger ADCP and ADNP effector functions in young children than in adults”, which is a strong overstatement given the minuscule differences shown. The significant differences they report are based on what appears to be 3.8 for adults vs 3.9 for vaccinated and 2.5 vs 2.7 (they don’t report median or range), but the data is clearly overlapping. Looking at Fig 4, where a very similar set of experiments are made, then the values are 3.6 for vaccinated and 3.7 for naturally infected children. Given the data they report themselves across one paper, it is quite clear that they don’t have any real differences, except potentially IgG3 in Fig. 4. But if that is also from 1 repeat, it may very well turn out to be no difference, once repeated.

Even though something might happen to be statistically significant, it doesn’t mean that the difference is large or important (p-hacking). Since the methodology used is not disclosed, nor are any method controls shown, it is impossible to say what a log value of 3.8 ADCP means. How many times were the experiments repeated? It looks like 1 time per figure.

5. The conclusions are not supported by the data since the authors extrapolate the importance of IgG3 in humoral immunity in the context of SARS-CoV-2. The importance of other classes, such as IgM and IgA is not mentioned, and the differences observed with IgG3 do not support the conclusion on vaccination vs natural infection. The authors mention that IgG3 is known to be a potent neutralizer, referring to several other work, and given the importance of neutralizing antibodies in humoral immunity and the way the authors talk about the importance of their findings, further experiments are warranted to test the neutralizing activity of the cohorts in this study. Vaccinated children, children with infection and the equivalent for adults. In addition, the lack of data on neutralizing activity further weakens the claims of the authors. Although IgG3 shows potent Fc-mediated function in vitro, its protective role in the context of SARS-CoV-2 infection is not clear so the authors extrapolation of their data is unjustified.

Minor points:

1. They write 8 children in the results text, yet have 5 children in the Table. I assume the former is the true value, since they show 8 data points in the figures. They need to add information on the additional 3 children in the Table.

2. Fig. 2. Multiple comparison tests should have been performed.

3. The authors stress the importance of IgG3s potent function in humoral immunity in line 243 onward and reference a few sources 16-18. Upon further reading, two of these references are reviews and not original research. There exist three other more suitable references that the authors should add which are of relevance to this subject. The work by Foss et al in context of Adenovirus and IgG3 (10.1126/sciimmunol.abj1640); the work on IgG3 neutralization against SARS-CoV-2 (<https://doi.org/10.1073/pnas.2107249118>) and by Izadi et al who shows potent fc-mediated function of IgG3 mAbs against Sars-cov-2 (<https://doi.org/10.1073/pnas.2217590120>).

Reviewer #3:

Remarks to the Author:

The manuscript investigates the humoral response of young children following vaccination against

SARS-CoV-2. The authors characterized antibody biophysical features and FcR mediated functions against different SARS-CoV-2 variants in three groups: vaccinated young children, naturally infected young children and vaccinated adults. They combined and analyzed the data using a systems serology approach.

They show that vaccination in young children induces a strong humoral response similar to that seen in adults. The antibodies produced against the SARS-CoV-2 RBD protein showed greater ADCP and ADNP functions among young children compared to adults. They also show that the antibody responses are cross-reactive between variants and that IgG3 response against Spike and RBD is higher post vaccination compared to natural infection.

The research is important to the area.

The results are straightforward to understand and its data assessment, interpretation and conclusions are all correct.

The specific conclusion that "vaccinated toddlers elicited a stronger functional antibody response than adults" would need to be more specific as the observation is only valid for ADCP and ADNP against RBD.

Some of the methods would also need to be more specific. Most likely some significant modifications were done to the methods from research articles cited in reference.

Minor comments

Line 83 and Line 521 (Table 1)

Could the age be expressed in year and months instead of decimals of years?

Line 83 and Line 521 (Table 1)

Age range in text is "7 months - 4.5 years" while in table it is "0.6, 4.5 years" In the table, none of the adults participants are represented.

Line 95

Add: "We saw that infants AND CHILDREN less than 5 years old..."

Line 111

Add:

"The breadth score highlights that INFANT AND children less than 5 years old..."

Line 122

According to the legend of Fig. S3, here "young children" refers to a different grouping than the infant and children less than 5 year old. Could it be specified here? Why was the grouping changed?

Line 127

"Co-correlates analysis showed strong connections between isotypes and FcγR features against non-SARS-CoV-2 antigens: What are the correlation coefficient values? The legend specifies the p values but not the coefficient values.

Line 128

"Fig. S1C" should be "Fig. S3C"

Line 130

"Alternatively, the lower antibodies could reflect...": the lower LEVEL of antibodies?

Line 142

Add: "...these infant AND CHILDREN"

Line 146

Replace the semi colon by a period: "Of note, IgM levels started to wane after V1 (Fig. 2A);"

Line 169-194

It is important to note that this is one month after infection only.

Was data collected from naturally infected children after one month from the onset of acute infection?

Line 246-250

It should be noted that the greater response from vaccinated children compared to naturally infected occurs at the 1 month time point.

Methods - Antibody isotype, FcR binding and functions (Lines 296-301)

Could you please describe the modifications to the assay compared to the method from reference 33, such as changes in the amount of antigen used, instrument, etc.?

Figure 1 A, B, C

What is the unit represented on the y-axis? Is it MFI, Fold over Background, or Baseline?

Figure 1 E legend (line 488)

"... responses for each secondary" - is it detection antibody? Can it be replace by "... responses for each immunoglobulin isotype and subclass"

Figure 2 legend (line 494)

Add: "... binding to FcγR (B) and function(C) against Spike WT was analyzed in INFANT AND children"

Figure 4

Legend has "γ" instead of "g" or "γ" for FcγR. Some of the text has the right annotation.

Supplemental Figure 1

For the children infected group, what is the estimated time (range) post infection?

Could all 3 graphs (IgG1, IgG3 and FcγR2a) be on the same scale?

Y axis title says "γ" instead of "g" or "γ" for FcγR. Some of the text has the right annotation.

Supplemental Figure 2

Why only 10 individuals in the child group when table reports n=19 for vaccinated children.

Supplemental Figure 3C

What is the difference between the orange and blue (FcγR2A_HKU1 - FcαR_OC43) lines? And the difference between the light green (IgG_HA) and dark green circles?

Dear reviewers,

Thank you for your valuable comments on the manuscript. Based on your remarks and suggestions, we have added enhanced details and descriptions in sections throughout the manuscript to enhance clarity or precision. We have expanded the discussion, and more particularly the methods section. In addition, we have modified the figures as requested. We believe that these changes have greatly improved the overall interpretability of the data and quality of the manuscript.

Below, we have included a point-by-point response to the different comments and concerns.

Reviewer 1

This manuscript by Nziza, Deng, Wood and colleagues describes the antibody profiles and function in young children less than 5 years of age following mRNA vaccination using systems serology. Responses were compared with adults as well as age-matched children infected with SARS-CoV-2. They report that children mount robust humoral antibody responses following vaccination (as expected) but that this response differed in some aspects when compared with adults or children who were infected. In particular, vaccinated children were found to have lower Spike and RBD-specific IgM and IgA1 responses compared with adults as well as stronger ADCP and ADNP effector functions. The authors also found some differences in the breadth of the response in children as well as responses to non-SARS-CoV-2 respiratory viruses compared with adults. Data was also presented over time (out to 6m following primary series, and following booster vaccination), showing waning of humoral immunity by 6 months but which was restored by the booster. Importantly, an IgG3 signature was identified in vaccinated children compared with infection, which provides novel evidence for the benefit of vaccinating children.

This is a very interesting manuscript that provides detailed information on the humoral immune response in young children following mRNA vaccination. This is the paper's strength since very limited data from young children have been reported in this context. I do however have a number of points that need to be considered by the authors.

Comment 1. The major limitation of this manuscript, as acknowledged by the authors, is the small samples size. Notwithstanding the variety of analyses conducted on these samples, the small numbers does make it difficult to draw meaningful conclusions. In particular, for some comparisons (e.g. timecourse Fig 2) there are <10 individuals. There are no sample size calculations provided which would provide confidence in these data.

Response:

We appreciate the reviewer's comment and acknowledge that a larger sample size would have been more advantageous. However, due to the limited vaccination of young infants – largely stemming from population hesitancy towards a novel vaccine technology, particularly for a young and fragile population – and the current reduced uptake of the COVID vaccine following the relaxation of restrictions, it has been arduous to include a high number of children under the age of 5 who were vaccinated with mRNA-1273 in this study. Several studies, facing similar challenges particularly with pediatric populations, have been conducted with small sample size (Indrayan and Mishra, 2021). They include projects on COVID-19 (Pierce et al., 2021), Respiratory Syncytial Virus (RSV) (Goodwin et al., 2018), or gene therapy for Artemis-deficient SCID (Cowan et al., 2022).

For this reason, existing published studies on covid vaccine and humoral immunity with large cohorts generally include adults and older children (Wei et al., 2021; Kaplonek et al., 2022a; Kaplonek et al., 2022b). However, it is interesting to note that our findings align with a previously published study indicating that children between 5 and 11 years old exhibit a more robust immune activation compared to adults (Bartsch et al., 2022). Overall, our study furnishes valuable insights into the characterization of humoral immunity in the younger pediatric population. Subsequent investigations are imperative to corroborate these findings across additional cohorts and various vaccination strategies, with the goal to unravel the mechanisms

underpinning immune responses within pediatric populations. This has been added to the discussion (lines 255-258, 270-271).

Comment 2. *Table 1 needs to include the number of samples studied at each timepoint as well as the time post vaccination or infection that the samples were measured (n in the text and on the figure legends). In the text, there are inconsistencies from 2-8 weeks post-infection and 1 month post. This could have an impact on the responses observed if only 2 weeks post-infection. If so, have the authors considered any analyses to account for time since infection/vaccination on the response observed?*

Response:

We apologize for the lack of precision. We have updated Table 1 and included the number of samples that were included for each timepoint. The numbers of samples are now also added to the legends. Additionally, we have corrected the section in the "Methods" referring to the number of weeks post-infection, which 5 weeks (average = 5.3 weeks \pm 2 weeks).

Comment 3. *A statement in the methods on the time period that these samples were collected. It appears that these were collected during the Delta wave but this is not made clear.*

Response:

This has been added to the manuscript, in the "Methods" section.

Comment 4. *For the cross-reactivity analyses with other respiratory infections (supplementary data), the authors report lower responses in children compared with adults to these antigens. This is consistent with the literature but given that the circulation of respiratory viruses were very low during the time that these samples were collected, could this be another explanation for the differences observed?*

Response:

We agree with the reviewer, and we added this comment to the manuscript, in the results, which now read "These antibody profiles in adults reflect prior exposure to these respiratory viruses over their lifetime, while these young children may remain naïve, particularly given the lower circulation of respiratory viruses observed during the COVID pandemic" (lines 134-135).

Comment 5. *The authors show data in young children <5years following a booster dose of mRNA vaccination. Is this an approved schedule in the US as I thought booster vaccinations were only recommended for children over 5 years of age? This needs to be clarified in the methods.*

Response:

Since October 2022, booster vaccination has been recommended by the CDC for patients younger than 5 years old, which allowed us to collect samples from these young infants. However, vaccine hesitancy in this young population remains strong, and this is the reason why only 3 post-boost samples were collected. This has been added to the manuscript (lines 270-271)

Comment 6. In some Figures, there is data on all IgG subclasses (Figs 1 and 4) but not others (Figs 2 and 3). Was this done for all analyses?

Response:

Yes, these analyses were done. They were initially not included to the manuscript as IgG2 and IgG4 are in low levels in the plasma, but we modified the figures in order to add these subclasses.

Comment 7. In Figure 1, it shows no difference in the FcγR response between children and adults, yet there was a difference in ADNP observed? What do the authors think might be the explanation for this?

Response:

The characterization of antibody FcR binding has been used to evaluate and predict antibody functionality (Bartsch et al., 2021). In our laboratory, we use recombinant FcR to analyze the capacity of antibodies to bind to FcR, and we use cells to characterize antibody dependent cellular phagocytosis (a description of this has been added to the “Methods” part). However, these antibody binding to recombinant FcR and cellular phagocytosis are not always directly correlated, as other factors, such as antibody glycosylation (Jennewein and Alter, 2017), can impact on the capacity of antibodies to bind to FcR located on cells. Other papers have shown results where no difference can be observed regarding FcR binding between two groups, when cellular phagocytosis differs between the same two populations (Nziza et al., 2023). This has been added to the manuscript (“Here we demonstrated that vaccinated young children display comparable Fc binding capacity as compared to vaccinated adults, while significantly higher antibody functionality was observed in the younger population in comparison to adults, showing a potential impact of age-dependent antibody glycosylation on the induction of phagocytosis”, lines 227-230).

Comment 8. Fig 4 legend states “..plasma of children 1 month after vaccination or diagnosis (on average)” – the authors should provide the precise numbers for this, ie mean and range or similar

Response:

We apologize for this lack of precision. The average is 5.3 weeks, \pm 2 weeks. This has been added to the manuscript (line 292).

Comment 9. For the breadth score (Fig 3), it states that a COVID-unexposed group was used to control for the response. Was this an age-matched control group? It is also not clear what the p-values are referring to on these graphs – are these global changes? It appears that this difference is largely attributed to the Omicron response which is not surprising particularly as this was mainly seen pre-booster.

Response:

The controls included in the breadth score analysis are COVID-unexposed adults, recruited from MGH with blood collected prior to the COVID-19 pandemic. This control, although not age matched, was the optimal, available sample population for assessment of assay background, as age-matched, pre-COVID pediatric samples were not available. Although it is conceivable that there may be some variation in background between pediatric and adult

populations in our assay, we prioritized comparing against pre-COVID serum rather than using a non-serum control, such as assay buffer. The p-values on the graph refer to global changes in two variables within the data set. Differences in variant responses were tested by mixed effects model with Geisser-Greenhouse correction with time and antibody features as the two effects that were modeled. For all comparisons between variants, there was a significant difference, and we agree that this is largely driven by lower omicron responses. Text explaining the control population and the statistical analysis has been added to the methods section of the paper.

Reviewer 2

They have performed a study of 19 mRNA-1273 vaccinated (2-doses) children under the age of 5 (25 mcg), and compared these to 8 naturally infected children. They also compared with an undisclosed group of 13 fully vaccinated (3-4 doses?) adults that had received 100 mcg of the vaccine. They find that the vaccinated children generated a robust IgG response, compared to adults, except for IgA and IgM levels, which are lower. They make strong statements about the benefits of vaccination compared to natural infection and that children have stronger functional responses than adults, which in my view, are not supported by the data shown.

The study has the potential to shed some light on the vaccination of young children, given that they have access to such samples, but the study design, experiments, and data presentation are severely lacking.

The results that appear to be clear are that young children get a good IgG response but weaker IgA and IgM response and that IgG3 levels might (pending on how the experiments were made, which is not clear) differ between vaccinated and infected children.

Major points:

Comment 1. Almost non-existent methods description. They very briefly refer to a Luminex-based method with no details. The reference they point to only studies subclass, and it would be impossible to reproduce subclass analysis given that they don't reveal any details. In addition, they state that they also measure IgA1, IgM, and FcyR2A, FcyR2B, FcyR3A, FcyR3B, and FcaR, none of which are described or measured in the cited reference.

Similarly, they report that they measure ADCP, ADCC, ADNP with no details. Only refer to other papers of the group, where it is impossible to find information specific to this study. Which cells do they use? At what concentration? Volume? Beads? Antigens? Time? The ratio of beads to cells? Titration curves? What controls did they use for the methods? etc. They write that they record MFI values but no flow cytometry details such as gatings, settings etc. How many times were the experiments repeated? This is really sub-standard and makes it impossible to assess the quality of the data.

Fig 1 and 4. ADCP, ADCC, ADNP log values of what? No description of what these values represent. One would have to guess. Which makes it impossible to evaluate. How many times were these analyses made? 1 time?

How were the heat maps constructed in Fig 1 and 4? From how many experiments?

Response:

We thank the reviewer for this comment and agree that the method section needed to be better described. We added details about the Luminex experiments, as well as the analysis of antibody functionality (with ADCD, ADCP and ADNP) (lines 332-358). We also included the gating strategy, which is now Fig. S6. For Fig. 1 and 4, it is the log₁₀ (MFI) values that are represented. We have updated the figures with clear unit to clarify this confusion. Concerning the heatmaps, they were constructed using the difference of the median Z-scored MFI data from two different groups. All experiments were performed in duplicate, and the average of the two duplicates is used for all the analysis. This has been added to the method part (lines 379-381).

Comment 2. In line 86-88 the authors stress that the anti-spike IgG levels are similar to adults. The comparison of the antibody response is to this adult cohort but the details concerning this cohort is limited. At what time point after vaccination were the IgG responses of the adult group compared? This needs to be clarified since it affects the interpretation (antibody titers wane over time) and can be a major bias in the interpretation of the results if the blood samples were taken several months after vaccination since it would affect titers and isotype presence etc.

The novelty would have been greatly improved if the analysis of the vaccinated and unvaccinated cohort would entail adults (at the same time point of 1 month) and include IgA and IgM as well as neutralization assays.

Response:

Thank you for noting this lack of clarity. In both groups, these samples were collected 2 months after the first dose of mRNA-1273 vaccine, which corresponds to 1 month after the second dose of the vaccine. This has been added to the results section, which reads: “In both groups, plasma samples were collected 2 months after the first vaccine dose, which corresponds to 1 month after the second dose of the vaccine” (lines 82-84).

In this project, samples from unvaccinated adults were not included, as previous papers describing the impact of COVID vaccines on antibody response in adults have been published by our team (Kaplonek et al., 2022a; Kaplonek et al., 2022b) and others (Wei et al., 2021). We have added these references to the manuscript, which now reads “with the novelty of the mRNA vaccine strategies, the impact on immune response in pediatric populations remains largely unknown, while several studies have described vaccine-dependent humoral activation in adults (Wei et al., 2021; Kaplonek et al., 2022a; Kaplonek et al., 2022b)”, (lines 204-205).

Moreover, concerning the study of antibody functionality, our goal was to focus on the non-neutralizing activity of antibodies, with the study of Fc-dependent antibody response, including ADCP, ADNP and ADCD. However, to answer to the reviewer’s comment, we performed neutralizing assays with vaccinated and infected children, as well as vaccinated adults and added the figure to the supplemental material. As represented on Fig. S3, antibody neutralization exhibited similar strength in both vaccinated adults and infected or vaccinated children. This has been added to the manuscript (lines 100-102).

Comment 3. The breadth score, found in graph 1E, shows that there is no difference between IgG3 levels for adults or children under 5. However, the authors show that natural infection vs vaccination, the humoral immune response differs for children (Figure 4). However, this same comparison with adults is not shown. This analysis for a comparable cohort (vaccinated adults one month after or infection one month after) is essential to show to observe if there is an age-dependent IgG3-skewed response depending on infection or vaccination.

Response:

We agree with the reviewer that the comparison of antibody response between infected and vaccinated adults is a crucial point to address in this project. Although plasma samples from adults infected with SARS-CoV-2 were not included in this study, as it has already been published that the antibody response is stronger after vaccination compared to infection in the adult population (Assis et al., 2021; Jalkanen et al., 2021; Kaplonek et al., 2022b; Meyers et al., 2022), these findings remain essential for the discussion. They have now been added to the manuscript (lines 255-258). Moreover, similar results have been described with older children, between 5 and 11 years old (Bartsch et al., 2022).

The conclusion we have from this study is that the differences we observe between vaccinated and infected children of a young age, more precisely less than 5 years old, mainly involve IgG3, which is interesting given the fact that IgG3 is the most functional antibody isotype, with binding efficiency to FcRs higher than IgG1 (Bruhns et al., 2009; Vidarsson et al., 2014; Kallolimath et al., 2021; Foss et al., 2022; Izadi et al., 2023). Whether the production of this highly functional isotype is associated with age-dependent mechanisms is unknown. Additional analyses with larger cohorts would be needed to further characterize the age-dependent impact of vaccination versus infection on antibody functionality, especially focusing on the study of antibody glycosylation profiles, which are known to play a key role in the modulation of antibody functionality. This has been added to the discussion (lines 261-264).

Comment 4. They describe “remarkably, anti-RBD antibodies exhibited stronger ADCP and ADNP effector functions in young children than in adults”, which is a strong overstatement given the minuscule differences shown. The significant differences they report are based on what appears to be 3.8 for adults vs 3.9 for vaccinated and 2.5 vs 2.7 (they don’t report median or range), but the data is clearly overlapping. Looking at Fig 4, where a very similar set of experiments are made, then the values are 3.6 for vaccinated and 3.7 for naturally infected children. Given the data they report themselves across one paper, it is quite clear that they don’t have any real differences, except potentially IgG3 in Fig. 4. But if that is also from 1 repeat, it may very well turn out to be no difference, once repeated (change axis).

Even though something might happen to be statistically significant, it doesn’t mean that the difference is large or important (p-hacking). Since the methodology used is not disclosed, nor are any method controls shown, it is impossible to say what a log value of 3.8 ADCP means. How many times were the experiments repeated? It looks like 1 time per figure.

Response:

We agree with the reviewer that the initial representation of the graphs was not convincing. The slight differences observed in the between adults and children resulted from the graphs' scale, which was not adapted for the analyses. Indeed, we used the same scale for all functions; however, it's important to note that phagosome values (used for ADCP and ADNP) are calculated differently from MFI values (used for ADCD). As a result, we generated new graphs using distinct scales. This adjustment allows us to observe that the overlap between adults and children is minimal for RBD-specific ADCP and ADNP.

All experiments were conducted twice, and the averages of the two replicates were used to construct the graphs. This information has been added to the “Methods” section (lines 379-381), along with a clearer description of the experiments and the controls used, as requested by the reviewer (lines 317-376).

Comment 5. The conclusions are not supported by the data since the authors extrapolate the importance of IgG3 in humoral immunity in the context of SARS-CoV-2. The importance of other classes, such as IgM and IgA is not mentioned, and the differences observed with IgG3 do not support the conclusion on vaccination vs natural infection. The authors mention that IgG3 is known to be a potent neutralizer, referring to several other work, and given the importance of neutralizing antibodies in humoral immunity and the way the authors talk about the importance of their findings, further experiments are warranted to test the neutralizing activity of the cohorts in this study. Vaccinated children, children with infection and the equivalent for adults. In addition, the lack of data on neutralizing activity further

weakens the claims of the authors. Although IgG3 shows potent Fc-mediated function in vitro, its protective role in the context of SARS-CoV-2 infection is not clear so the authors extrapolation of their data is unjustified.

Response:

Considering the rapid waning of IgM following vaccination due to class switch recombination (Stavnezer et al., 2008; Narasimhan et al., 2021; Assaid et al., 2023), and with IgA being a mucosal isotype (Pabst and Slack, 2020; Russell and Mestecky, 2022), our interest was more focused on IgG levels, which is the major isotype in the blood (Vidarsson et al., 2014). However, in response to the reviewer's comment, we have added these results to the supplemental material, in addition to neutralizing activity, as explained above (Fig. S3 and Fig. S4).

Also, here, we aim to show that IgG3 against all the VOCs, which is the most functional isotype in the blood (Bruhns et al., 2009; Vidarsson et al., 2014; Kallolimath et al., 2021; Foss et al., 2022; Izadi et al., 2023), is higher after vaccination compared to vaccination. However, we agree with the reviewer that further analyses are required to validate this IgG3-specific functionality. A more in-depth characterization of the IgG3 profile, particularly regarding glycosylation, within this young cohort will bring a deeper understanding of the pathways involved in the immune activation in this young population.

Minor points:

Comment 1. They write 8 children in the results text, yet have 5 children in the Table. I assume the former is the true value, since they show 8 data points in the figures. They need to add information on the additional 3 children in the Table.

Response:

We are grateful for the reviewer's noting this error. This has been corrected and the Table 1 now shows 8 children.

Comment 2. Fig. 2. Multiple comparison tests should have been performed.

Response:

We thank the reviewer for this important comment, new graphs have been added, using Benjamini-Hochberg correction for multiple testing. This has been added to the legend of Fig. 2.

Comment 3. The authors stress the importance of IgG3s potent function in humoral immunity in line 243 onward and reference a few sources 16-18. Upon further reading, two of these references are reviews and not original research. There exist three other more suitable references that the authors should add which are of relevance to this subject. The work by Foss et al in context of Adenovirus and IgG3 (10.1126/sciimmunol.abj1640); the work on IgG3 neutralization against SARS-CoV-2 (<https://doi.org/10.1073/pnas.2107249118>) and by Izadi et al who shows potent fc-mediated function of IgG3 mAbs against Sars-cov-2 (<https://doi.org/10.1073/pnas.2217590120>).

Response:

We appreciate this helpful feedback. These references have been added to the manuscript.

Reviewer 3

The manuscript investigates the humoral response of young children following vaccination against SARS-CoV-2. The authors characterized antibody biophysical features and FcR mediated functions against different SARS-CoV-2 variants in three groups: vaccinated young children, naturally infected young children and vaccinated adults. They combined and analyzed the data using a systems serology approach.

They show that vaccination in young children induces a strong humoral response similar to that seen in adults. The antibodies produced against the SARS-CoV-2 RBD protein showed greater ADCP and ADNP functions among young children compared to adults. They also show that the antibody responses are cross-reactive between variants and that IgG3 response against Spike and RBD is higher post vaccination compared to natural infection.

The research is important to the area.

The results are straightforward to understand and its data assessment, interpretation and conclusions are all correct.

Major comments:

Comment 1. The specific conclusion that “vaccinated toddlers elicited a stronger functional antibody response than adults” would need to be more specific as the observation is only valid for ADCP and ADNP against RBD.

Response:

We thank the reviewer for this comment and acknowledge the lack of precision. This sentence has been changed and now reads “Despite their lower vaccine dose, vaccinated toddlers elicited a functional antibody response as strong as adults, with higher antibody-dependent phagocytosis compared to adults, without report of side effects”, lines 40-42.

Comment 2. Some of the methods would also need to be more specific. Most likely some significant modification were done to the methods from research articles cited in reference.

Response:

We acknowledge the reviewer's feedback and extend our apologies for the insufficient level of detail in the "Methods" section. To address this concern, we have incorporated further explanations into both the Luminex experiments and the functional analyses (detailed in lines 317-376).

Minor comments:

Comment 1. Line 83 and Line 521 (Table 1)

Could the age be expressed in year and months instead of decimals of years?

Response:

We thank the reviewer for pointing out the inconsistency in the unit of age, with months in the text, and decimals of years in the table. This has been corrected, and we are now using decimals to maintain coherence and consistency with previous papers published by the team (Bartsch et al., 2022) (line 80).

Comment 2. Line 83 and Line 521 (Table 1)

Age range in text is “7 months - 4.5 years” while in table it is “0.6, 4.5 years” In the table, none of the adults participants are represented.

Response:

This has been corrected; decimal years are now used in both the table and the text. Additionally, we appreciate the reviewer for pointing out the absence of information on the adult cohort. This cohort was included and described in another paper (Ogata et al., 2022), which is now referenced in our manuscript (line 295).

Comment 3. Line 95

Add: “We saw that infants AND CHILDREN less than 5 years old...”

Response:

These changes have been applied.

Comment 4. Line 111

Add:

“The breadth score highlights that INFANT AND children less than 5 years old...”

Response:

These changes have been applied.

Comment 5. Line 122

According to the legend of Fig. S3, here “young children” refers to a different grouping than the infant and children less than 5 year old. Could it be specified here? Why was the grouping changed?

Response:

We are grateful for the reviewer’s noting this error. We used the same group of infants and children less than 5 years old, therefore we corrected the legend and wrote “5 years old” instead of “4 years old”.

Comment 6. Line 127

“Co-correlates analysis showed strong connections between isotypes and FcγR features against non-SARS-CoV-2 antigens: What are the correlation coefficient values? The legend specifies the p values but not the coefficient values.

Response:

We presented correlations with p-value < 0.01 and the absolute value of correlation coefficient > 0.8. This has been added to the legend.

Comment 7. Line 128

“Fig. S1C” should be “Fig. S3C”

Response:

This sentence has been changed.

Comment 8. Line 130

“Alternatively, the lower antibodies could reflect...”: the lower LEVEL of antibodies?

Response:

This sentence has been corrected.

Comment 9. Line 142

Add: “..these infant AND CHILDREN”

Response:

This sentence has been changed.

Comment 10. Line 146

Replace the semi colon by a period: “Of note, IgM levels started to wane after V1 (Fig. 2A);”

Response:

This change has been done.

Comment 11. Line 169-194

It is important to note that this is one month after infection only.

Was data collected from naturally infected children after one month from the onset of acute infection?

Response:

These data were collected after one month from diagnosis, as mentioned on the “Methods” section. This has now also been added to the “Results” section (line 179).

Comment 12. Line 246-250

It should be noted that the greater response from vaccinated children compared to naturally infected occurs at the 1 month time point.

Response:

This has been added to the text.

Comment 13. Methods - Antibody isotype, FcR binding and functions (Lines 296-301)

Could you please describe the modifications to the assay compared to the method from reference 33, such as changes in the amount of antigen used, instrument, etc.?

Response:

We agree with the reviewer, and we apologize for the lack of clarity. The “Methods” section has been modified and a more detailed description of the techniques has been added (lines 317-376).

Comment 14. Figure 1 A, B, C

What is the unit represented on the y-axis? Is it MFI, Fold over Background, or Baseline?

Response:

We thank the reviewer for this helpful question. Those units on the y-axis represent raw MFI. We have updated the figures to reflect this.

Comment 15. Figure 1 E legend (line 488)

“... responses for each secondary” - is it detection antibody? Can it be replace by “... responses for each immunoglobulin isotype and subclass”

Response:

We agree with the reviewer, and we have changed that sentence.

Comment 16. Figure 2 legend (line 494)

Add: “... binding to FcyR (B) and functon(C) against Spike WT was analyzed in INFANT AND children”

Response:

This sentence has been corrected.

Comment 17. Figure 4

Legend has “y” instead of “g” or “γ” for FcgR. Some of the text has the right annotation.

Response:

We are thankful to the reviewer for noting this inconsistency. This has been corrected, and γ is now used in the text and on the figures.

Comment 18. Supplemental Figure 1

For the children infected group, what is the estimated time (range) post infection?

Could all 3 graphs (IgG1, IgG3 and FcgR2a) be on the same scale?

Y axis title says “y” instead of “g” or “γ” for FcgR. Some of the text has the right annotation.

Response:

The average and estimated range are 5.3 weeks \pm 2 weeks. This has been added to the “Methods” section (line 292).

Given the fact that we used different sample dilutions for the different isotypes and FcR binding results (this has now been added to the method section), a direct comparison between these different features cannot be done. This is the reason why different scales were used, which allows us to better visualize antibody response over time.

The mistakes concerning the FcR notation have now been corrected.

Comment 19. Supplemental Figure 2

Why only 10 individuals in the child group when table reports n=19 for vaccinated children.

Response:

Fig. S2 only includes the timepoint corresponding to 2 months post vaccination, which is the reason why all 19 vaccinated children are not represented. This has been added to the legend, as well as in Table 1.

Comment 20. Supplemental Figure 3C

What is the difference between the orange and blue (FcgR2A_HKU1 - FcaR_OC43) lines?

And the difference between the light green (IgG_HA) and dark green circles?

Response:

The orange lines represent positive correlation and blue lines are negative correlation. The light green dots represent the features not selected by LASSO and the dark green circles are the LASSO selected features.

References

- Assaid, N., Arich, S., Charoute, H., Akarid, K., Anouar Sadat, M., Maaroufi, A., et al. (2023). Kinetics of SARS-CoV-2 IgM and IgG Antibodies 3 Months after COVID-19 Onset in Moroccan Patients. *Am J Trop Med Hyg* 108(1), 145-154. doi: 10.4269/ajtmh.22-0448.
- Assis, R., Jain, A., Nakajima, R., Jasinskas, A., Khan, S., Palma, A., et al. (2021). Distinct SARS-CoV-2 antibody reactivity patterns elicited by natural infection and mRNA vaccination. *NPJ Vaccines* 6(1), 132. doi: 10.1038/s41541-021-00396-3.
- Bartsch, Y.C., St Denis, K.J., Kaplonek, P., Kang, J., Lam, E.C., Burns, M.D., et al. (2022). SARS-CoV-2 mRNA vaccination elicits robust antibody responses in children. *Sci Transl Med* 14(672), eabn9237. doi: 10.1126/scitranslmed.abn9237.
- Bartsch, Y.C., Wang, C., Zohar, T., Fischinger, S., Atyeo, C., Burke, J.S., et al. (2021). Humoral signatures of protective and pathological SARS-CoV-2 infection in children. *Nat Med* 27(3), 454-462. doi: 10.1038/s41591-021-01263-3.
- Bruhns, P., Iannascoli, B., England, P., Mancardi, D.A., Fernandez, N., Jorieux, S., et al. (2009). Specificity and affinity of human Fcγ receptors and their polymorphic variants for human IgG subclasses. *Blood* 113(16), 3716-3725. doi: 10.1182/blood-2008-09-179754.
- Cowan, M.J., Yu, J., Facchino, J., Fraser-Browne, C., Sanford, U., Kawahara, M., et al. (2022). Lentiviral Gene Therapy for Artemis-Deficient SCID. *N Engl J Med* 387(25), 2344-2355. doi: 10.1056/NEJMoa2206575.
- Foss, S., Jonsson, A., Bottermann, M., Watkinson, R., Lode, H.E., McAdam, M.B., et al. (2022). Potent TRIM21 and complement-dependent intracellular antiviral immunity requires the IgG3 hinge. *Sci Immunol* 7(70), eabj1640. doi: 10.1126/sciimmunol.abj1640.
- Goodwin, E., Gilman, M.S.A., Wrapp, D., Chen, M., Ngwuta, J.O., Moin, S.M., et al. (2018). Infants Infected with Respiratory Syncytial Virus Generate Potent Neutralizing Antibodies that Lack Somatic Hypermutation. *Immunity* 48(2), 339-349 e335. doi: 10.1016/j.immuni.2018.01.005.
- Indrayan, A., and Mishra, A. (2021). The importance of small samples in medical research. *J Postgrad Med* 67(4), 219-223. doi: 10.4103/jpgm.JPGM_230_21.
- Izadi, A., Hailu, A., Godzwon, M., Wrighton, S., Olofsson, B., Schmidt, T., et al. (2023). Subclass-switched anti-spike IgG3 oligoclonal cocktails strongly enhance Fc-mediated opsonization. *Proc Natl Acad Sci U S A* 120(15), e2217590120. doi: 10.1073/pnas.2217590120.
- Jalkanen, P., Kolehmainen, P., Hakkinen, H.K., Huttunen, M., Tahtinen, P.A., Lundberg, R., et al. (2021). COVID-19 mRNA vaccine induced antibody responses against three SARS-CoV-2 variants. *Nat Commun* 12(1), 3991. doi: 10.1038/s41467-021-24285-4.
- Jennewein, M.F., and Alter, G. (2017). The Immunoregulatory Roles of Antibody Glycosylation. *Trends Immunol* 38(5), 358-372. doi: 10.1016/j.it.2017.02.004.
- Kallolimath, S., Sun, L., Palt, R., Stiasny, K., Mayrhofer, P., Gruber, C., et al. (2021). Highly active engineered IgG3 antibodies against SARS-CoV-2. *Proc Natl Acad Sci U S A* 118(42). doi: 10.1073/pnas.2107249118.
- Kaplonek, P., Cizmeci, D., Fischinger, S., Collier, A.R., Suscovich, T., Linde, C., et al. (2022a). mRNA-1273 and BNT162b2 COVID-19 vaccines elicit antibodies with differences in Fc-mediated effector functions. *Sci Transl Med* 14(645), eabm2311. doi: 10.1126/scitranslmed.abm2311.

- Kaplonek, P., Fischinger, S., Cizmeci, D., Bartsch, Y.C., Kang, J., Burke, J.S., et al. (2022b). mRNA-1273 vaccine-induced antibodies maintain Fc effector functions across SARS-CoV-2 variants of concern. *Immunity* 55(2), 355-365 e354. doi: 10.1016/j.immuni.2022.01.001.
- Meyers, J., Windau, A., Schmotzer, C., Saade, E., Noguez, J., Stempak, L., et al. (2022). SARS-CoV-2 antibody profile of naturally infected and vaccinated individuals detected using qualitative, semi-quantitative and multiplex immunoassays. *Diagn Microbiol Infect Dis* 104(4), 115803. doi: 10.1016/j.diagmicrobio.2022.115803.
- Narasimhan, M., Mahimainathan, L., Noh, J., and Muthukumar, A. (2021). Silent SARS-CoV-2 Infections, Waning Immunity, Serology Testing, and COVID-19 Vaccination: A Perspective. *Front Immunol* 12, 730404. doi: 10.3389/fimmu.2021.730404.
- Nziza, N., Tran, T.M., DeRiso, E.A., Dolatshahi, S., Herman, J.D., de Lacerda, L., et al. (2023). Accumulation of Neutrophil Phagocytic Antibody Features Tracks With Naturally Acquired Immunity Against Malaria in Children. *J Infect Dis*. doi: 10.1093/infdis/jiad115.
- Ogata, A.F., Cheng, C.A., Desjardins, M., Senussi, Y., Sherman, A.C., Powell, M., et al. (2022). Circulating Severe Acute Respiratory Syndrome Coronavirus 2 (SARS-CoV-2) Vaccine Antigen Detected in the Plasma of mRNA-1273 Vaccine Recipients. *Clin Infect Dis* 74(4), 715-718. doi: 10.1093/cid/ciab465.
- Pabst, O., and Slack, E. (2020). IgA and the intestinal microbiota: the importance of being specific. *Mucosal Immunol* 13(1), 12-21. doi: 10.1038/s41385-019-0227-4.
- Pierce, C.A., Sy, S., Galen, B., Goldstein, D.Y., Orner, E., Keller, M.J., et al. (2021). Natural mucosal barriers and COVID-19 in children. *JCI Insight* 6(9). doi: 10.1172/jci.insight.148694.
- Russell, M.W., and Mestecky, J. (2022). Mucosal immunity: The missing link in comprehending SARS-CoV-2 infection and transmission. *Front Immunol* 13, 957107. doi: 10.3389/fimmu.2022.957107.
- Stavnezer, J., Guikema, J.E., and Schrader, C.E. (2008). Mechanism and regulation of class switch recombination. *Annu Rev Immunol* 26, 261-292. doi: 10.1146/annurev.immunol.26.021607.090248.
- Vidarsson, G., Dekkers, G., and Rispens, T. (2014). IgG subclasses and allotypes: from structure to effector functions. *Front Immunol* 5, 520. doi: 10.3389/fimmu.2014.00520.
- Wei, J., Stoesser, N., Matthews, P.C., Ayoubkhani, D., Studley, R., Bell, I., et al. (2021). Antibody responses to SARS-CoV-2 vaccines in 45,965 adults from the general population of the United Kingdom. *Nat Microbiol* 6(9), 1140-1149. doi: 10.1038/s41564-021-00947-3.

Reviewers' Comments:

Reviewer #1:

Remarks to the Author:

Thankyou to the authors for addressing the points raised by this reviewer. This has improved the quality of the manuscript.

Reviewer #2:

Remarks to the Author:

I commend the authors for addressing most of my concerns with the manuscript. The only remaining minor issue is related to my previous comment 5, reproduced here with their response:

Comment 5. The conclusions are not supported by the data since the authors extrapolate the importance of IgG3 in humoral immunity in the context of SARS-CoV-2. The importance of other classes, such as IgM and IgA is not mentioned, and the differences observed with IgG3 do not support the conclusion on vaccination vs natural infection. The authors mention that IgG3 is known to be a potent neutralizer, referring to several other work, and given the importance of neutralizing antibodies in humoral immunity and the way the authors talk about the importance of their findings, further experiments are warranted to test the neutralizing activity of the cohorts in this study. Vaccinated children, children with infection and the equivalent for adults. In addition, the lack of data on neutralizing activity further 9 weakens the claims of the authors. Although IgG3 shows potent Fc-mediated function in vitro, its protective role in the context of SARS-CoV-2 infection is not clear so the authors extrapolation of their data is unjustified.

Author Response: Considering the rapid waning of IgM following vaccination due to class switch recombination (Stavnezer et al., 2008; Narasimhan et al., 2021; Assaid et al., 2023), and with IgA being a mucosal isotype (Pabst and Slack, 2020; Russell and Mestecky, 2022), our interest was more focused on IgG levels, which is the major isotype in the blood (Vidarsson et al., 2014). However, in response to the reviewer's comment, we have added these results to the supplemental material, in addition to neutralizing activity, as explained above (Fig. S3 and Fig. S4). Also, here, we aim to show that IgG3 against all the VOCs, which is the most functional isotype in the blood (Bruhns et al., 2009; Vidarsson et al., 2014; Kallolimath et al., 2021; Foss et al., 2022; Izadi et al., 2023), is higher after vaccination compared to vaccination. However, we agree with the reviewer that further analyses are required to validate this IgG3-specific functionality. A more in-depth characterization of the IgG3 profile, particularly regarding glycosylation, within this young cohort will bring a deeper understanding of the pathways involved in the immune activation in this young population.

Reviewer response to revised manuscript:

As discussed above the importance of antibody functionality depends on several factors, of which the authors now have included neutralizing ability of their cohort. However, I have an issue remaining with the following statements in lines 253-255 where the authors state that their data suggests that mRNA vaccination grants superior protection due to higher IgG3 titers. That statement is not supported due to the absence of clinical data to support such a claim. My previous point on extrapolating conclusions based on in vitro data alone remains.

To elaborate: even if IgG3 is the most functional IgG subclass in blood, having higher levels and function in vitro does not necessarily correspond to higher in vivo function. Even if the authors had performed in vivo experiments (K18-hACE2 mice infected with VOCs, for instance) with the patient plasma (containing these more potent IgG3 titers) and seen similar trends following the ADCP and ADCC data it is a large leap to state that these experimental data correlates to clinical relevance (without large data set from clinical settings to correlate ADCC/ADCP functionality to clinical outcome). Pharmacokinetic aspects such as lower IgG3 half-life would influence outcome in vivo. The authors

mention IgG3 glycosylation as another important aspect, but so is IgG3 stability in blood and half-life as well, which are important aspects to discuss.

I strongly suggest that the authors discuss these important limitations of their study in terms of interpreting the clinical relevance.

Thank you for the reviewers' valuable comments on the manuscript. We have revised the manuscript to address the follow up concern (detailed below). Thank you for the thoughtful feedback and we are thrilled that our manuscript has been accepted for publication.

Reviewer 2

Reviewer response to revised manuscript:

As discussed above the importance of antibody functionality depends on several factors, of which the authors now have included neutralizing ability of their cohort. However, I have an issue remaining with the following statements in lines 253-255 where the authors state that their data suggests that mRNA vaccination grants superior protection due to higher IgG3 titers. That statement is not supported due to the absence of clinical data to support such a claim. My previous point on extrapolating conclusions based on in vitro data alone remains.

To elaborate: even if IgG3 is the most functional IgG subclass in blood, having higher levels and function in vitro does not necessarily correspond to higher in vivo function. Even if the authors had performed in vivo experiments (K18-hACE2 mice infected with VOCs, for instance) with the patient plasma (containing these more potent IgG3 titers) and seen similar trends following the ADCP and ADCC data it is a large leap to state that these experimental data correlates to clinical relevance (without large data set from clinical settings to correlate ADCC/ADCP functionality to clinical outcome). Pharmacokinetic aspects such as lower IgG3 half-life would influence outcome in vivo. The authors mention IgG3 glycosylation as another important aspect, but so is IgG3 stability in blood and half-life as well, which are important aspects to discuss.

I strongly suggest that the authors discuss these important limitations of their study in terms of interpreting the clinical relevance.

Response:

We thank the reviewer for these clarifications and apologize the lack of precision in our response. We agree that we are not comparing clinical manifestations between COVID infection and vaccination, therefore our sentence that mentions superior protection might be confusing.

We have removed that sentence and further elaborated this limitation in the discussion, as requested by the reviewer. We wrote, “Moreover, to address the question regarding which group is associated with superior clinical protection against COVID-19, further studies involving pharmacokinetic aspects, such as IgG3 half-life and the stability of this isotype in the blood of this pediatric population, as well as correlations between clinical features and antibody functionality, would be valuable in determining whether mRNA vaccination confers superior protection than SARS-CoV-2 infection.”

The new modifications are highlighted in blue.